# Sensitivity and Calibration of the FT-IR Spectroscopy on Concentration of Heavy Metal Ions in River and Borehole Water Sources

**Matthew Mamera \*** , **Johan J. van Tol** , **Makhosazana P. Aghoghovwia** and **Elmarie Kotze**

Faculty of Natural Sciences, Department of Soil, Crop and Climate Sciences, University of the Free State, Bloemfontein 9301, South Africa; vantoljj@ufs.ac.za (J.J.v.T.); AghoghovwiaMP@ufs.ac.za (M.P.A.); KotzeE@ufs.ac.za (E.K.)

**\*** Correspondence: 2015319532@ufs4life.ac.za; Tel.: +27-67-090-4564

**Abstract:** Heavy metals in water sources can threaten human life and the environment. The analysis time, need for chemical reagents, and sample amount per analysis assist in monitoring contaminants. Application of the Fourier Transform Infrared (FT-IR) Spectroscopy for the investigation of heavy metal elements has significantly developed due to its cost effectiveness and accuracy. Use of chemometric models such as Partial Least Square (PLS) and Principle Component Regression Analysis (PCA) relate the multiple spectral intensities from numerous calibration samples to the recognized analytes. This study focused on the FT-IR calibration and quantification of heavy metals (Ag, Cd, Cu, Pb and Zn) in surveyed water sources. FT-IR measurements were compared with the atomic absorption spectrometer (AAS) measurements. Quantitative analysis methods, PCA and PLS, were used in the FT-IR calibration. The spectral analyses were done using the Attenuated Total Reflectance (ATR-FTIR) technique on three river and four borehole water sources sampled within two seasons in QwaQwa, South Africa (SA). The PLS models had good $R^2$ values ranging from 0.95 to 1 and the PCA models ranged from 0.98 to 0.99. Significant differences were seen at 0.001 and 0.05 levels between the PLS and PCA models for detecting Cd and Pb in the water samples. The PCA models detected Ag concentrations more (<0 mg L$^{-1}$ on selected sites). Both the PLS and PCA models had lower detection only for Zn ions mostly above 45 mg L$^{-1}$ deviating from the AAS measurements (<0.020 mg L$^{-1}$). The FT-IR spectroscopy demonstrated good potential for heavy metal determination purposes.

**Keywords:** FT-IR calibration; chemometric models; heavy metals; water resource contamination

## 1. Introduction

The scale of contaminant migration from sanitation systems highly depends on variable climatic conditions, range of geological formations and soilscapes [1]. Heavy metals are non-biodegradable and persist in the aqueous solution as toxic substances within the environment [2]. The release of heavy metal wastes towards water resources can cause many physical, chemical and biological modifications [3]. Accumulation of heavy metals in water sources cause threats to human life and the environment [2]. Heavy metal level determination in water resources is important, since most of them are harmful even at low concentrations [3]. Heavy metals can be a challenge in catchment water resources for an extended time, even when the source has been remediated [2]. In most rural and some urban areas in developing countries, people rely solely on untreated surface and groundwater sources for drinking purposes [1]. Monitoring, on a regular basis, of these water resources can sometimes be costly to meet the recommended guidelines for drinking water sources [1,4,5]. Heavy metal ions in solution tend to be difficult to remove with the use of traditional techniques [2,6]. Frequently applied

techniques for measuring heavy metal concentration, such as inductively coupled plasma optical emission spectrometry (ICP-OES), are time-consuming and expensive [2,7,8]. The analysis time, cost, laboratorial simplicity, need for chemical reagents, and sample amount per analysis can be essential in monitoring water contaminants [9].

The Fourier Transform Infrared (FT-IR) Spectroscopy has emerged to be very effective in analysis of any sample [9–12]. Infrared spectroscopy has proven to have potential in the analysis and concentration of heavy metal contamination [2,6,8]. Application of the FT-IR techniques for the investigation of heavy metal elements has been significantly developed due to their flexibility and accuracy [7,8,13,14]. Advances in the IR spectroscopy has led to the use of an interferometer and the Fourier-transform algorithm. The FT-IR spectroscopy operates through the interface of an IR beam with a sample. IR radiation excites the vibration of the coordination bond in metal complexes, the ionic bond in crystals and the covalent bonds between atoms. The product is an infrared spectrum that mirrors the complete chemical composition of the sample [9].

FT-IR is a non-destructive method normally used to distinguish the functional groups responsible for heavy metals adsorption [13,14]. The applied FT-IR method can also be useful for characterization of heavy metals in ionic or sub-nanosized cluster form in a colloidal dispersed system and metals dispersed in solution [15]. The infrared position can give information for oxidation state, bonding and morphology of the dispersed clusters of heavy metal ions [16,17]. To attain reproducible results, sample treatment procedures, IR measurement techniques and IR acquisition parameters must be controlled and calibrated rigidly [10]. Use of chemometric models such as Partial Least Square (PLS) and Principle Component Regression Analysis (PCA) relate the multiple spectral intensities from numerous calibration samples to the recognized analytes within samples by removing the random variation (noise) and retaining the principal components that capture the important variation [15,18].

Nonetheless, the challenge on the choice of the optimal measurement procedure remains imprecise [7]. There is also a critical question on the dynamic issues of representativeness of samples for spectral analyses and relationship of the measurements in the laboratory after sample preparation or direct analysis. Moreover, other concerns relate to the choice of spectral methods, the influence of the difference of the material studied on the results, or the choice of chemometric techniques to create a calibration and prediction model [8,19,20].

Here we hypothesize that the selected heavy metals of Silver (Ag), Cadmium (Cd), Copper (Cu), Lead (Pb) and Zinc (Zn) in solution can be efficiently identified from water sources with the proper calibration of the FT-IR spectrometer, thus being cost effective for sample analysis. The objective of this paper was to establish the best fitting calibration curves using known concentrations of metal standards; then secondly to verify the sensitivity between the PLS and PCA quantifying methods using river and borehole water quality samples; and lastly, to determine the accuracy of the FT-IR when compared to an Atomic Absorption Spectrometer (AAS).

## 2. Materials and Methods

In this study, standard grade reagents purchased from Sigma Aldrich were used to prepare known concentration (i.e., 0, 2.5, 5, 10, 15 and 30 mg $L^{-1}$ of Ag, Cd, Cu, Pb and Zn) calibration solutions. Each respective heavy metal standard used in the calibration models was replicated 5 times. The salts used were lead nitrate ($Pb(NO_3)_2$) for preparing lead; cadmium sulphate ($3CdSO_4, 8H_2O$) for cadmium; copper sulfate ($CuSO_4, 5H_2O$) for copper; zinc sulphate ($ZnSO_4, 7H_2O$) for zinc; and silver nitrate ($AgNO_3$) for silver.

A Thermo Scientific Nicolet iS50 FT-IR spectrometer was used to measure the standards for the absorbance calibration methods. The spectral analyses were done using the Attenuated Total Reflectance (ATR-FTIR) technique. Each metal ion calibration model was constructed from 35 spectral scans. A total of 175 spectral scans were run for all the models. Approximately a milliliter of each of the heavy metal solutions was pipetted and placed on the built-in diamond iS50 ATR sampling station. This direct measurement technique enables propagation of the infrared radiation to pass at an angle

that allows internal reflectance from the crystal. The crystal was cleaned with alcohol (70% Acetone) and de-ionized water following each successive spectrum sampling. Prior to analysis, the spectrum for air was used as a background in the technique. In the study, spectra wavelength was obtained for the mid-infrared range (400–4000 cm$^{-1}$). The relationship of the heavy metal concentrations and the spectral properties of the solutions was obtained with multivariate calibration methods. The spectral data were calibrated with TQ-Analyst 9 software (Thermo Fisher Scientific Inc., Madison, WI, USA, 2017). Quantitative analysis methods of Principal Component Analysis (PCA) and Partial Least Square (PLS) were used. The data was transformed by smoothing using the Savitzky-Golay filter algorithm with a third order polynomial.

PLS and PCR methods were preferred due to their ability to use the intensities of all data points within specified regions in the analysis [21–23]. These techniques should include at least three calibration standards for each component being measured and additional validation standards. Although the two methods are similar, the PCR method can also use multiple regions of the spectrum. In addition, all of the components in a PCR method can be measured in each region. The PLS method is capable of quantifying sample components when the correlation between concentration and absorbance is very complex. Furthermore, overlapping component peaks can be reduced with the PLS method. It is more suitable in cases where chemical interactions between components cause peaks in the mixture spectrum to shift or broaden. It is also useful where additional components whose concentrations are unknown may be present in the sample mixture [21–23].

The PCA and PLS models were determined based on similar Cross-Validation (CV) following the methods by Sliwinska et al., [8] and Wold, [24]. The correct PCA and PLS models were characterized with a goodness-of-fit to the data used in the construction and to ensure good prediction capability for new variables. Linearity of the calibration models was expressed by the coefficient of determination ($R^2$) and goodness of fit [25]. If $R^2$ is closer to or further from 1, the correlation is higher or lower, respectively. In this study, the use of Root Mean Square Error (RMSE) (Equation (1)) and the root square error were applied in cross validations (RMSCV) (Equation (2)), using the formulas [8,24]:

$$RMS = \sqrt{\frac{\sum_{i=1}^{n} (y_i - \hat{y}_{\cdot i})^2}{n}} \tag{1}$$

where $y_i$ and $\hat{y}_{\cdot i}$ stand for the experimental measurements for the dependent variable in the models and the expected measurements, respectively, and n stands for the total variables in the models;

$$RMSCV\ (C) = \sqrt{\frac{\sum_{i=1}^{n} \left(y_i^t - \hat{y}_{\cdot i}^t(C)\right)^2}{n}} \tag{2}$$

where; $y_i^t$ and $\hat{y}_{\cdot i}^t$ (C) stand for measurements obtained in the test set and their measurements generated by the model for the complexity A, correspondingly.

Sensitivity and accuracy were done using field survey water quality samples from QwaQwa, eastern Free State province, South Africa within two seasons (i.e., October to December 2019 and January to February 2020). In the survey, four rivers (QM 2, 4, 5 and 6) and three boreholes (QM 3, 7 and 8) sources were analyzed with an AAS at the Institute of Ground Water Studies (IGS), University of the Free State (UFS) laboratory and compared with the Thermo Scientific Nicolet iS50 FT-IR spectrometer measurements.

The water samples data was subjected to independent sample t-tests using XLStats version 365.1 statistical software to check the differences between PLS and PCA methods with respect to heavy metal detections. Significant differences were considered at a 0.1% and 5% probability level.

*Ethical Approval*

The field survey for the water quality monitoring in this study was approved by the Environmental and Biosafety Research Ethics Committee (EBREC) No. UFS-ESD2019/0066.

## 3. Results

### 3.1. FT-IR Spectra of Water Containing Metal Ions

In the study, major peaks were observed at wavenumber ranges of 2845 to 3698 cm$^{-1}$ and 1557 to 1720 cm$^{-1}$ (Figure 1). The main bands observed were mostly due to the water medium, in addition to the metal ions affecting the water bands, resulting in stretching and bending of bonds. The initial IR spectra of the metals that were used in the standards and analyzed in the water samples are represented with the following colors for all the spectra: silver (purple), cadmium (green), copper (light blue), lead (pink), zinc (red) and de-ionized water, representing the blank (dark blue).

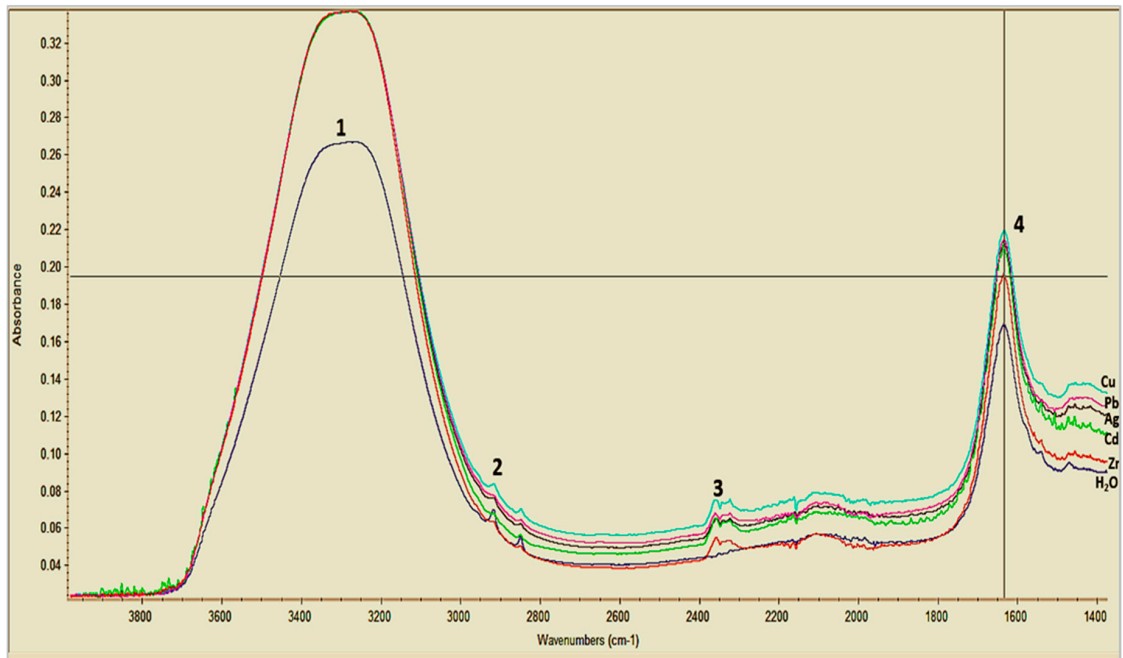

**Figure 1.** Showing the Fourier Transform Infrared (FT-IR) spectra metal ions in the standards and the water samples.

Absorbance of the blank samples was observed at 0.26. All the metal ions (Ag, Cd, Cu, Pb and Zn) caused a change in the water band at an absorbance of 0.34 (Figure 1). The blank sample peaks were used as reference points for all metal ion detections in this study (Table 1). The peak positions with heavy metals (Figure 1 and Table 1) were measured using the region, spectral cursor and peak height tools in TQ analyst software. In the first peak, slight stretching shifts were assigned to Ag at a band of 3357 cm$^{-1}$ and Cd (3353 cm$^{-1}$). A decrease in the peak wavenumbers occurred in the spectra for Cu assigned at a band of 3349 cm$^{-1}$, Pb (3330 cm$^{-1}$) and Zn (3341 cm$^{-1}$).

**Table 1.** Bands width and the peak positions in the Attenuated Total Reflectance (ATR)/FT-IR spectra.

| | Peak Position without Heavy Metals (cm$^{-1}$) | | Peak Position with Heavy Metals (cm$^{-1}$) | | | | |
|---|---|---|---|---|---|---|---|
| | Assignment | Blank | Ag | Cd | Cu | Pb | Zn |
| 1 | O-H Stretching; Symmetric stretches | 3351–3252 | 3357–3246 | 3353–3255 | 3349–3248 | 3330–3264 | 3341–3264 |
| 2 | O-H Stretching | 2921 | 2917 | 2917 | 2914 | 2915 | 2916 |
| 3 | Symmetrical bending; O-H Stretching | 2353 | 2359 | 2361 | 2365 | 2363 | 2361 |
| 4 | Symmetrical bending | 1637 | 1637 | 1635 | 1637 | 1635 | 1637 |

In the second peak, all the water band spectra stretching frequencies were assigned to bands in the region 2921–2914 cm$^{-1}$. There was a decrease in the water bands with the presence of all metal ions. In the third peak, an increase occurred in the water band shifting the wavenumbers. The water band was assigned to 2353 cm$^{-1}$, and all the added metal ions were assigned to bands in the region 2365–2359 cm$^{-1}$. In the fourth peak of the water band, changes were only observed in the bending motion of the water band with the presences of Cd and Pb assigned at a band of 1635 cm$^{-1}$ as contrasted to the blank samples at a wavenumber of 1637 cm$^{-1}$ (Table 1).

## 3.2. Calibration Models

The PLS and PCA methods calculated concentration values well in the models. This was verified when the data points in the calculated vs. actual concentration plot formed a line exactly 45 degrees from both axes Figures 2 and 3). The PCA calibration absorbance models had good RMSEC correlation coefficients with R$^2$ values of 0.98 (Ag, Cu, Pb) and 0.99 (Cd, Zn) (Figure 2a–e). The Ag model had the highest RMSEC value of 2.43, while the lowest was observed in Zn models with a value of 1.48.

The PLS calibration absorbance models had R$^2$ values of 0.95 (Ag), 0.98 (Cu) and 1 (Cd, Pb, Zn) (Figure 3). There were differences in the R$^2$ values for Ag, Cu, Pb and Zn between the PCA models (Figure 2) and PLS models (Figure 3). The only similarity was in the models for Cu, Figures 1c and 2c. Slightly lower RMSEC values were observed in the PLS as compared to PCA models. In these models, the highest RMSEC was also observed in the Ag model with a value of 3.42. The lowest was obtained in Pb models with a value of 0.55. Models for Pb had a higher prediction ability, while Ag had the lowest. Generally, all the methods had good prediction abilities as the RMSEC were mostly below 2.50, irrespective of the metal ion.

In the presented results, the model correction was based on calibration values (Figures 2 and 3). Correction, cross-correction and ignored standards were only used in the pre-processing of the data for the method corrections. Each model in Figures 2 and 3 used 10 standards for data validation in the pre-processing for the presented calibration models.

Data normalization in the models based on the mean centering technique determined the number of principal components used in the PLS and PCR models as shown in Table 2. This resulted in each generated model (Figures 2 and 3) using different principal component factors (Table 2) to increase the prediction of variables.

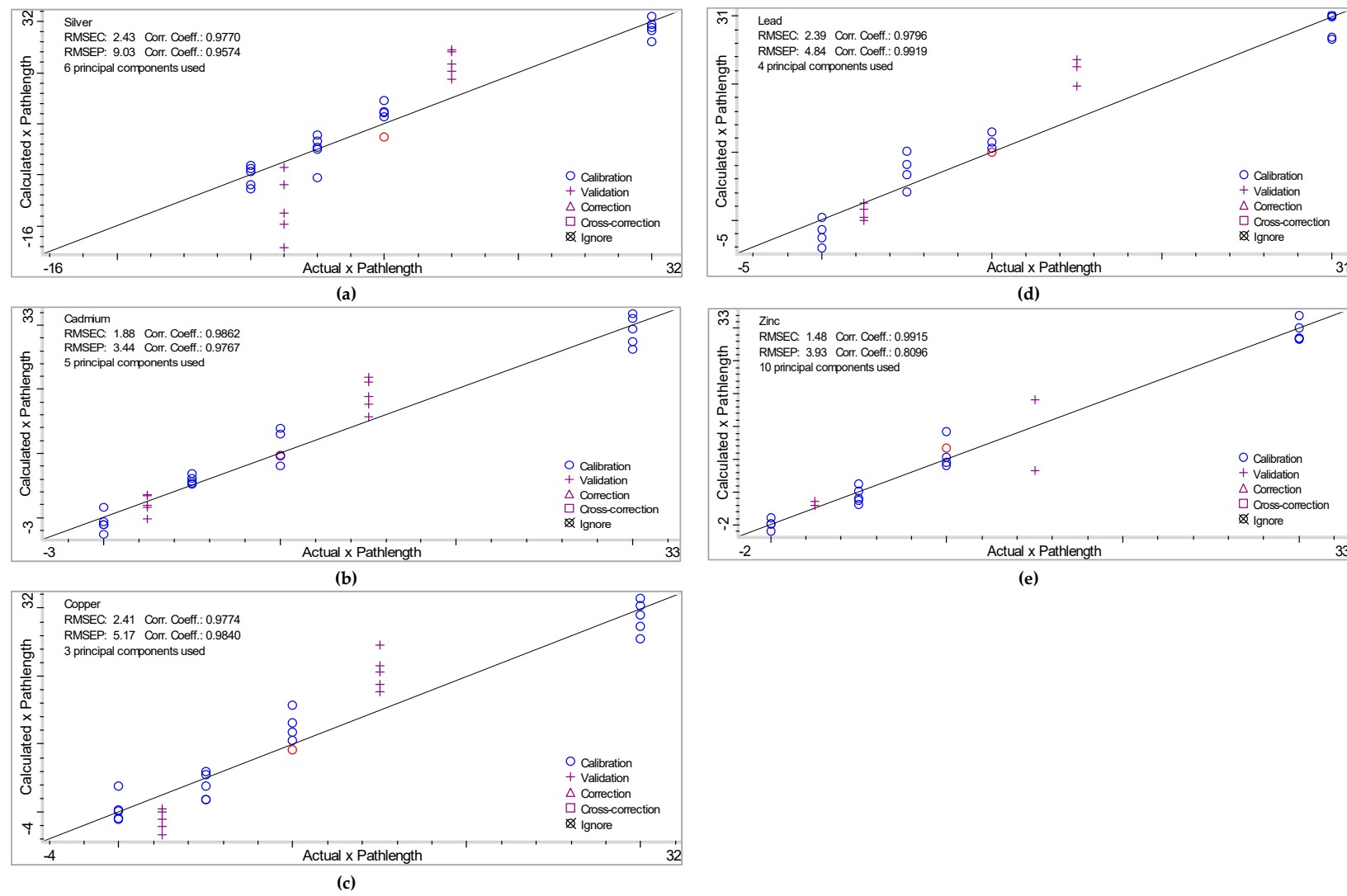

**Figure 2.** Showing the PCA calibration models for caption (**a**) Ag; (**b**) Cd; (**c**) Cu; (**d**) Pb; and (**e**) Zn.

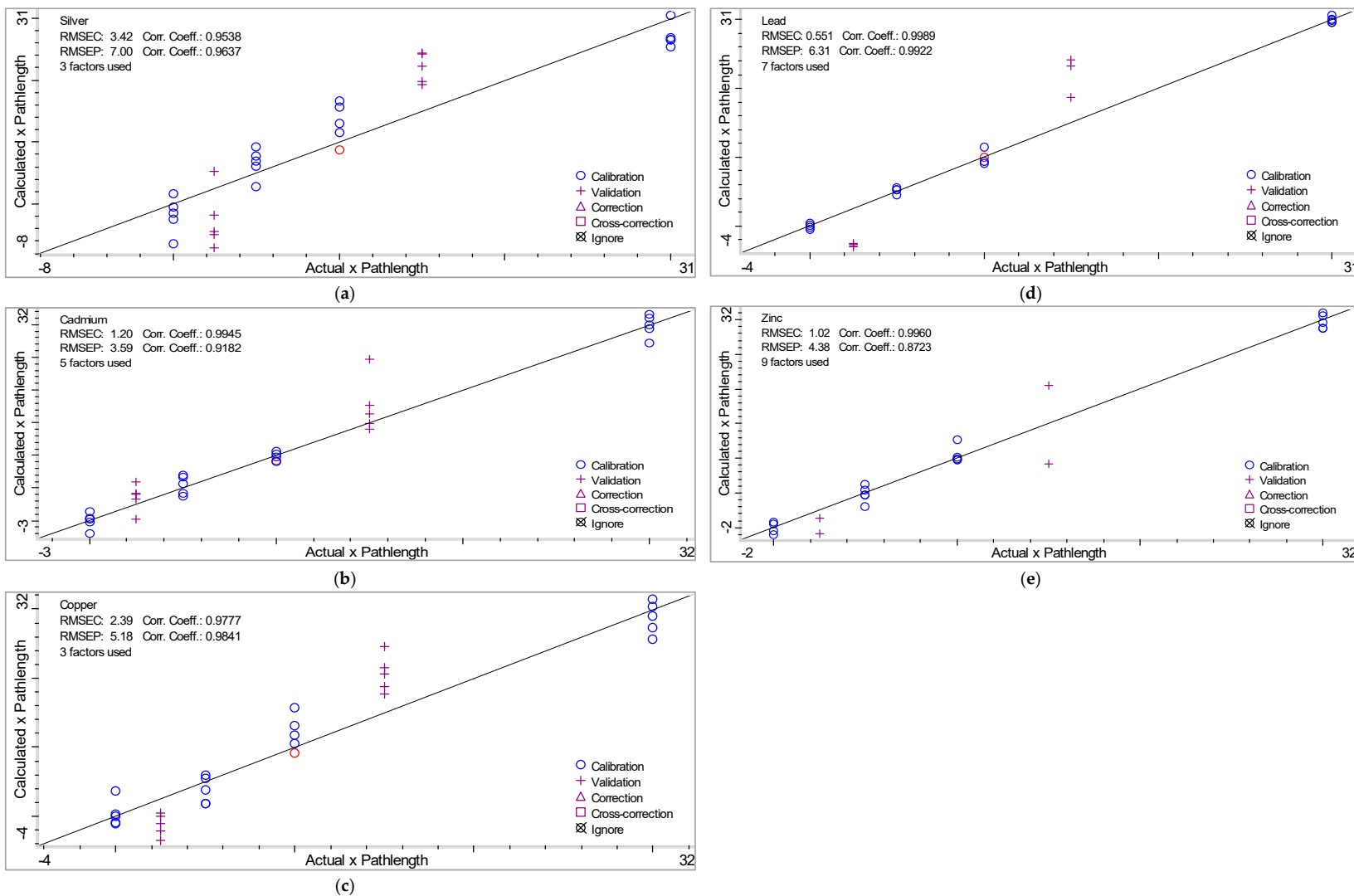

**Figure 3.** Showing the PLS calibration models for caption (**a**) Ag; (**b**) Cd; (**c**) Cu; (**d**) Pb; and (**e**) Zn.

**Table 2.** Principal component values for the Partial Least Square (PLS) and Principal Component Analysis (PCA) models based on ATR FT-IR data output.

| | Number of Model Variables | | | | |
|---|---|---|---|---|---|
| Metal | Ag | Cd | Cu | Pb | Zn |
| PLS | 6 | 5 | 3 | 4 | 10 |
| PCA | 3 | 5 | 3 | 7 | 9 |

### 3.3. Heavy Metal Detection Using the PLS and PCA Methods

Heavy metal concentrations observed using the PLS technique had lower values as compared to the PCA technique (Table 3). The PLS and PCA methods showed a significant difference on both 0.001 and 0.05 probability levels in quantifying mostly Cd and Pb in the water samples. There were no significant differences in the methods on the detection of Ag, Cu and Zn. Concentrations above 45 mg $L^{-1}$ for Zn were obtained between October to December 2019. Even though Zn values declined in the successive sampling period, detections were still high in the water samples. The PLS and PCA methods detected the same concentrations in borehole water sources, below 0 mg $L^{-1}$ for Ag, Cd, Cu and Pb (QM 8) and Cu, Pb (QM 5 and QM 7). In the study, mostly the PCA method showed concentrations below 0 mg $L^{-1}$ for Ag values in both river and borehole sources. The only exception was in site QM 8, where the PLS method also detected values below 0 mg $L^{-1}$ Ag. On the other hand, the PLS method strongly detected concentrations below 0 mg $L^{-1}$ for Cd and Pb.

**Table 3.** Comparison of heavy metal concentration (mg L$^{-1}$) in water sources as determined with the PLS and PCA methods.

| | | | October–December 2019 | | | | | January–February 2020 | | | | |
| | | | PLS | | PCA | | | PLS | | PCA | | |
| Source | Site | Element | Mean | SD | Mean | SD | t-Test | Mean | SD | Mean | SD | t-Test |
|---|---|---|---|---|---|---|---|---|---|---|---|---|
| River | | Ag | 29.27 | 1.29 | 26.76 | 0.13 | 0.11 | 2.20 | 3.81 | 0 | 0 | 0.37 |
| | | Cd | 0 | 0 | 27.42 | 1.92 | 0.002 * | 0 | 0 | 3.31 | 3.33 | 0.16 |
| | QM-2 | Cu | 17.75 | 0.30 | 17.79 | 0.31 | 0.91 | 1.33 | 2.30 | 1.36 | 2.36 | 0.99 |
| | | Pb | 3.22 | 3.54 | 4.59 | 2.79 | 0.71 | 0 | 0 | 0.13 | 0.23 | 0.37 |
| | | Zn | 69.54 | 10.15 | 74.00 | 13.29 | 0.74 | 9.34 | 8.20 | 14.19 | 14.25 | 0.64 |
| | | Ag | 17.83 | 0.42 | 15.48 | 0.98 | 0.09 | 20.59 | 3.00 | 21.50 | 2.64 | 0.71 |
| | | Cd | 0 | 0 | 19.50 | 1.45 | 0.003 * | 0 | 0 | 2.64 | 24.17 | 0.00 ** |
| | QM-4 | Cu | 10.31 | 0.21 | 10.31 | 0.21 | 0.98 | 14.26 | 2.29 | 14.32 | 2.29 | 0.97 |
| | | Pb | 0.05 | 0.07 | 0 | 0 | 0.42 | 0 | 0 | 1.77 | 3.07 | 0.37 |
| | | Zn | 50.97 | 4.89 | 59.63 | 2.47 | 0.16 | 8.50 | 8.27 | 11.73 | 10.91 | 0.70 |
| | | Ag | 2.36 | 1.11 | 0 | 0 | 0.10 | 32.20 | 2.24 | 31.10 | 7.20 | 0.81 |
| | | Cd | 0 | 0 | 5.64 | 0.76 | 0.01 * | 0 | 0 | 35.16 | 2.60 | 0.00 ** |
| | QM-5 | Cu | 0 | 0 | 0 | 0 | 1.00 | 21.70 | 1.60 | 21.76 | 1.64 | 0.96 |
| | | Pb | 0 | 0 | 0 | 0 | 1.00 | 0 | 0 | 6.51 | 1.18 | 0.001 ** |
| | | Zn | 54.76 | 9.33 | 63.18 | 8.13 | 0.44 | 31.31 | 13.09 | 31.86 | 10.28 | 0.96 |
| | | Ag | 44.49 | 0.35 | 48.93 | 2.26 | 0.11 | 4.03 | 2.75 | 5.81 | 1.98 | 0.41 |
| | | Cd | 0 | 0 | 37.16 | 1.89 | 0.001 ** | 0 | 0 | 4.93 | 3.57 | 0.08 |
| | QM-6 | Cu | 28.24 | 1.30 | 28.34 | 1.31 | 0.95 | 0.52 | 0.89 | 0.51 | 0.88 | 0.99 |
| | | Pb | 5.64 | 2.12 | 22.20 | 1.39 | 0.01 * | 0 | 0 | 0 | 0 | 1.00 |
| | | Zn | 66.25 | 13.72 | 66.94 | 11.24 | 0.96 | 17.54 | 19.00 | 22.54 | 21.27 | 0.78 |
| | | Ag | 17.59 | 4.65 | 8.29 | 2.82 | 0.14 | 8.89 | 5.99 | 0 | 0 | 0.06 |
| | | Cd | 0 | 0 | 15.32 | 3.08 | 0.02 * | 0 | 0 | 10.34 | 3.47 | 0.01 * |
| | QM-3 | Cu | 10.42 | 4.37 | 10.45 | 4.41 | 1.00 | 2.27 | 2.50 | 2.48 | 2.27 | 0.92 |
| | | Pb | 0.48 | 0.68 | 5.69 | 8.05 | 0.46 | 0 | 0 | 0 | 0 | 1.00 |
| | | Zn | 57.89 | 7.42 | 59.58 | 1.08 | 0.78 | 19.80 | 12.28 | 25.72 | 4.41 | 0.48 |
| Borehole | | Ag | 1.24 | 1.75 | 0 | 0 | 0.42 | 16.06 | 5.79 | 8.85 | 5.78 | 0.20 |
| | | Cd | 0 | 0 | 2.29 | 0.43 | 0.02 * | 0 | 0 | 17.33 | 4.22 | 0.002v * |
| | QM-7 | Cu | 0 | 0 | 0 | 0 | 1.00 | 9.83 | 3.79 | 9.86 | 3.82 | 0.99 |
| | | Pb | 0 | 0 | 0 | 0 | 1.00 | 0 | 0 | 4.89 | 4.16 | 0.11 |
| | | Zn | 45.67 | 10.32 | 51.08 | 5.06 | 0.57 | 15.66 | 27.12 | 19.79 | 26.86 | 0.86 |
| | | Ag | 0 | 0 | 0 | 0 | 1.00 | 25.56 | 3.08 | 20.47 | 3.82 | 0.15 |
| | | Cd | 0 | 0 | 0 | 0 | 1.00 | 0 | 0 | 27.98 | 1.37 | 0.00 ** |
| | QM-8 | Cu | 0 | 0 | 0 | 0 | 1.00 | 13.39 | 0.41 | 13.45 | 0.38 | 0.87 |
| | | Pb | 0 | 0 | 0 | 0 | 1.00 | - | - | 0.87 | 1.51 | 0.37 |
| | | Zn | 50.05 | 4.05 | 55.22 | 1.56 | 0.23 | 20.94 | 19.01 | 21.16 | 26.88 | 0.99 |

SD-Standard deviation of means; * and ** Significance at 0.05 and 0.001 probability level.

### 3.4. Heavy Metals Chemical Analysis (AAS)

Generally, all study sites for river and borehole sources had concentrations below 0.2 mg L$^{-1}$ for the analyzed heavy metals (Table 4). The chemical results (Table 4) indicated that similar values below 0 mg L$^{-1}$ were obtained mostly for Cd and Pb using the PLS method (Table 3).

**Table 4.** Chemical analysis of heavy metals in water samples.

| Source | Site | Element | October–December 2019 Value (mg L$^{-1}$) | January–February 2020 Value (mg L$^{-1}$) |
|---|---|---|---|---|
| River | QM-2 | Ag | <0.020 | <0.020 ** |
| | | Cd | <0.003 * | <0.003 * |
| | | Cu | 0.018 | 0.008 |
| | | Pb | <0.015 | <0.015 * |
| | | Zn | <0.020 | <0.020 |
| | QM-4 | Ag | <0.020 | <0.020 |
| | | Cd | <0.003 * | <0.020 * |
| | | Cu | 0.017 | <0.003 |
| | | Pb | <0.015 *** | 0.008* |
| | | Zn | <0.020 | <0.015 |
| | QM-5 | Ag | <0.020 ** | <0.020 |
| | | Cd | <0.003 * | <0.003 * |
| | | Cu | 0.015 *** | 0.013 |
| | | Pb | <0.015 *** | <0.015 * |
| | | Zn | <0.020 | <0.020 |
| | QM-6 | Ag | <0.020 | <0.020 |
| | | Cd | <0.003 * | <0.003 * |
| | | Cu | 0.020 | 0.011 |
| | | Pb | <0.015 | <0.015 *** |
| | | Zn | 0.023 | <0.020 |
| Borehole | QM-3 | Ag | <0.020 | <0.020 ** |
| | | Cd | <0.003 * | <0.003 * |
| | | Cu | 0.014 | 0.005 |
| | | Pb | <0.015 * | <0.015 *** |
| | | Zn | <0.020 | <0.020 |
| | QM-7 | Ag | <0.020 ** | <0.020 |
| | | Cd | <0.003 * | <0.003 * |
| | | Cu | 0.012 *** | <0.005 |
| | | Pb | <0.015 *** | <0.015 * |
| | | Zn | 0.129 | 0.217 |
| | QM-8 | Ag | <0.020 *** | <0.020 |
| | | Cd | <0.003 *** | <0.003 * |
| | | Cu | 0.015 *** | <0.005 |
| | | Pb | <0.015 *** | <0.015 |
| | | Zn | 0.060 | <0.020 |

Results similar to the AAS concentrations: * PLS method; ** PCA method; *** both methods.

Concentrations of Ag (0 mg L$^{-1}$) using the PCA method were also similar in both water sources to the AAS values (<0.020 mg L$^{-1}$). Both the PLS and PCA methods were able to detect Cu (0 mg L$^{-1}$) similar to the chemical results (0.020 mg L$^{-1}$). Even though the concentrations were slightly high using the AAS as contrasted to the FT-IR, they were mostly within acceptable threshold levels for national recommended standards in drinking water sources (i.e., Ag $\leq$ 0.05 mg L$^{-1}$; Cd $\leq$ 0.003 mg L$^{-1}$; Cu $\leq$ 2 mg L$^{-1}$; Zn- $\leq$ 5 mg L$^{-1}$); WHO, [4] and SANS, [5]. The only expectation was for Pb, which was slightly higher (0.015 mg L$^{-1}$) than the threshold levels (Pb $\leq$ 0.01 mg L$^{-1}$). Both the PLS and PCA methods detected higher Zn values (Table 3) than the actual concentrations (<0.060 mg L$^{-1}$) obtained from an AAS analysis (Table 4).

## 4. Discussion

IR spectra peaks of Cu, Pb and Cd similar to Figure 1 and Table 1 have been reported in other studies [26]. Spectra interpretations are important when developing chemometric models [18]. The PLS and PCA analysis methods are useful when each measured component produces a measurable peak in the spectrum of the sample mixture [21–23]. This was also observed in this study in Figure 1. Metals can influence the IR spectra of solutions and obtained spectral variations can be valuable to determine metal detections with the use of chemometric approaches [8]. Complexes containing oxygen with metal ions cause frequency shifting and changes in the intensity and shape of bands. This is usually applied in the analysis of metal concentrations studies with an FT-IR. In this study, Figure 1 and Table 1 show that a similar effect could have influenced the spectra shifts due the presence of oxygen ions in water. The effect of metal bonding in organometallic compounds loosens the bonds causing lowering of the wavelength in a spectra value, more than is seen in free groups [8]. Findings on a study using waste waters suggested that metal ion shifts in a spectrum can be associated with O-H, C-H, C=C, C=O and C-O bonds movement in the functional groups [26]. Results of this study also compare well to Karthikeyan, [3], in a study where heavy metals were induced in water. Their findings suggest that shifts between 3301 to 3281 cm$^{-1}$ are assigned to O-H stretching, 2960 to 2874 cm$^{-1}$ to CH$_3$ and CH$_2$ symmetric stretching and 1664 to 1652 cm$^{-1}$, inplane C = O stretching. However, in the present study the solutions only had water and metal ions, which is unlikely to cause CH$_3$ and CH$_2$ symmetric stretching and C=O stretching. Studies have showed that the IR spectra of water contains three bands as predicted by the Group Theory [27–29]. The modes of vibration in water bands are related to dipole moment which are also infra-red active [29,30]. Water has two stretching vibrations and one bending vibration. Two of the symmetric stretches occur at 3756 and 3657 cm$^{-1}$, whereas the bending motion occurs at 1595 cm$^{-1}$ [27,30,31]. This might also have been the cause in the stretching of the water bands in Figure 1 and Table 1 due to the effect of the studied metal ions. In the present study, closer water band ranges were obtained from all induced metal ions (3357 to 1635 cm$^{-1}$) in the respective peaks in Figure 1.

Use of PLS and PCA models have been implemented in heavy metal studies based on their regression fit and prediction abilities [8,20]. In the present study, higher correlations (R$^2$) from 0.95 to 1 were observed in the models in Figures 2 and 3. Studies have shown that the FT-IR spectroscopy is an acceptable method to assess heavy metal ions abundance and adsorption [20,26]. Studies suggest that if the R$^2$ value is lower than 0.8, the linearity and prediction value have a low correlation with the actual value and the prediction is unfeasible [23]. Furthermore, the closer the RSMEC value is to 0, the higher the prediction of a model [23,32]. In the present study, models in Figures 2 and 3 had similar low RSMEC values. Yongliang and Hee-Jin [33] in their study successfully reported R values which presented strong and linear relations using FT-IR calibration techniques. Findings in Figures 2d,e and 3d,e in this study compared well with the observations by Sliwinska et al., [8] as zinc and lead had a relatively good fit in their calibration models. However, their results were different on the copper models which presented a weak fit and prediction of the element. In this study, higher linear correlations values were obtained for both PLS and PCA. This result confirmed that rigid cross validation of the sample in standard calibration could have improved the sensitivity of the prediction of the models. This can also be seen on the higher detection of most metals in the study of the PLS calibration method. A study by Naumann [10] also suggested that a controlled and vigorous calibration of the FT-IR increases the ability to quantify and identify an unknown sample. Studies have also revealed that PCA methods require more calibration standards to increase its prediction ability [10,18,20]. This could have been the reason the PCA as compared to the PLS model results in Table 3 were mostly higher (for all the heavy metals except Ag). Moreover, in cases where regions in a spectrum do not contain absorptions of the analyzed components, better results are obtained by using the PLS method as it is more suitable in quantifying components in water [21–23].

Matwijczuk et al., [19]; Yongliang and Hee-Jin, [33] and Wang et al., [34] also used an ATR-FTIR spectroscopy and their studies had good regression models and high correct identification rates of

samples with the PCA method. This might have been the reason that the PCA method (Table 2), although it had some limitations (i.e., for Cd, Cu, Pb and Zn), was more capable of detecting Ag with values similar to the AAS results (Table 4) as compared to the PLS method. Studies have also shown that $Ag^+$ ions are sorbed in the highest amount in soils and hence detect low concentrations in water sources [26]. Silver ions have the shortest ionic radii of all studied cations, which can influence their highest sorption rate. Instead, $Pb^{2+}$ and $Ag^+$ ions are sorbed better on soils [34]. These effects were observed in our study because the results showed that heavy metal concentrations in groundwater sources and surface water sources were similar to boreholes and rivers, respectively.

Tajuddin et al. [26] and Njoki et al. [2], in studies for heavy metal adsorption in solution, were able to detect Cd, Cu and Pb with the FT-IR spectroscopy. Their findings on Cd, Cu and Pb also agreed with the results obtained in this study. Analytical measurement techniques generally are closer in determination of most chemical elements [19,33–35]. However, as in the present study, some mean deviations might also occur based on several factors. Kenawy et al. [35], in a study comparing the AAS and the FT-IR of metal ions (Ag, Cd, Co, Cr, Cu, Fe, Mn, Ni, Pb, Pd and Zn) in tap, Nile, waste, sea water samples, human urine and milk, suggested that interfering effects of different foreign ions can cause measurement variation from different analytical methods. In the present study, this might also have been the cause in some of the deviations which were seen between the PLS and PCA values (Table 3) and AAS values in Table 4. Their findings showed that variations in the detection of the metal ions between the different analytical methods were influenced by the function of pH, mass of ion exchanger at equilibrium, cations and anions in solution. Similar to this study, high levels of Zn ions above 84 $\mu$ g $L^{-1}$ were also observed as compared to all their studied metal ions with the FT-IR compared to the AAS with no detection. Cd, Ag and Pd ions were not detected in their tap water with the AAS as compared to other sources [35]. In this study, the methods used detected these metals using the FT-IR and AAS techniques. Gonzalez-Albarran et al. [36], in a study measuring Cadmium (II) in tap, bottle and pier water samples, also had comparable results between the new MID-FTIR-PLS PIM based-sensor and F-AAS analytical methods. Other studies argue that analytical procedures for determining heavy metal ions can be strongly interfered with matrix constituents such as alkali and alkaline earth elements [35–39]. In this present study, groundwater sources similar to boreholes can also contain soluble salts in solution. The underlying geological substrata can contribute to the concentration of other interfering elements. Studies have shown that interfering anions such as acetate, oxalate, citrate, $SO_4^{2-}$, $NO_3$-, $PO_4^{3-}$, $K^+$ and $H_2Y^{2-}$ can cause deviations in the concentration of heavy metals among measurement methods similar to the AAS and FT-IR [35,36]. Results of a spectroscopic study on a natural clinoptilolite (a zeolite) comparing the FT-IR and AAS for heavy metal cations ($Pb^{2+}$, $Cd^{2+}$, $Ag^+$ and $Cr^{3+}$) observed good metal ion detection and sorption in solution [25]. Lead and cadmium ions were sorbed with comparable ion-exchange and chemisorption process proportions [2,40]. The amount of lead detectable in aqueous forms is comparable with the cadmium sorption via ion-exchange process. The total amount of $Pb^{2+}$ in water sources is significantly higher than $Cd^{2+}$ due to domination of lead chemisorption process over cadmium. Similar findings were also observed in results for both the PLS and PCA methods [34].

Individual metals form complexes with organic groups of different energy [8]. Metal bonds formed in aqueous forms have different bond energies [2]. The stronger the new bond is with the metal, the weaker the organic group becomes, which results in a wider frequency of the spectra peak for the pure organic group [8]. Such an effect also influences the solution abundance of heavy metals. This may have been the reason higher concentrations of zinc were observed. Higher values than the recommended threshold levels for heavy metals concentration in water sources have also been observed in studies using the FT-IR [41]. Studies have also proposed that interference can also occur in identifying a sample, especially in a water-based media due to the effect of bonding energies. This might also have been the cause for higher values observed for most metals in Table 3 [9]. Njoki et al. [2] reviewed that chelating and complexation are possible reactions in the adsorption process which determines availability of metal ions in solution. Moreover, metal ions can also be adsorbed by substitution

reactions or by electrostatic forces of attraction to the electron cloud [2]. The FT-IR procedure allows aqueous analysis of the sorption/desorption phenomena which can assist in determination of the speciation of sorbed inorganic anions or inorganic complexes formed [42]. The results obtained in this study can also be beneficial in establishing the transport model of toxic species in natural waters and remediation of liquid wastes [11,42,43].

*Implications*

Even though calibration and prediction challenges using chemometric methods in the FT-IR spectroscopy have been reported [7,43], its potential could be implemented in the monitoring of water quality changes over time for heavy metal concentrations. Moreover, its efficiency can assist in the mapping of pollute levels and migration across large areas for water resources within catchments. In most developing countries, like the monitored sites in this study, water pollution challenges can be persistent because of the available sanitation systems and waste management [1,44–46]. Guidelines in water resource protection recommends regular monitoring of water quality [4,5]. Improvements in accuracy of chemometric methods such PLS and PCA using the FT-IR spectroscopy can also enable sustainable options among the existing analysis techniques. Its use can promote rapid assessment [43,47–51] in contaminated soil and water resources. In the present study, selected metals had higher detections depending on the chemometric method applied, indicating a higher potential of the FT-IR use.

## 5. Conclusions

Four major peaks were observed in the FT-IR spectra bands of the water samples. All heavy metals, Ag, Cd, Cu, Pb and Zn, affected the water bands to shift within a wavenumber range of 3357 to 1635 cm$^{-1}$. Calibration models for PLS and PCA methods had good regression fits for Ag, Cd, Cu, Pb and Zn ions. The PLS models had $R^2$ values ranging from 0.95 to 1 and the PCA models ranged from 0.98 to 0.99. PLS models predicted Cd and Pb (less than 0 mg L$^{-1}$) in both the river and borehole sources well. Cu was the least detected heavy metal using the PLS models. Significant differences were seen at 0.001% and 0.05% between the PLS and PCA models on detecting only Cd and Pb in the water samples. The PCA models highly detected Ag concentrations (less than 0 mg L$^{-1}$ on selected sites) as compared to the other heavy metals. The PLS had a higher sensitivity and accuracy to detect heavy metals in solution (below 0 mg L$^{-1}$) when validated with the chemical results (AAS analysis) below 0.2 mg L$^{-1}$ for all the elements. Both the PLS and PCA models detected high Zn ions (mostly above 45 mg L$^{-1}$) in the water samples (River and Borehole). However, the FT-IR spectroscopy demonstrated good potential and can be reliable for heavy metal determination purposes. Rigid and controlled calibrations can reduce costs for water resource monitoring and analysis. To increase its reliability, it is recommended to increase the number of standard spectra scans when constructing calibration models, especially for the PCA method. Future work should aim to further evaluate and verify the accuracy of these multi-component methods using different aqueous solutions as well. Another focus should also further compare the PLS and PCA to other well-known analytical methods such as the Atomic Absorption Spectrometer (AAS).

**Author Contributions:** Conceptualization, M.M., J.J.v.T and M.P.A.; methodology, M.M., J.J.v.T and E.K.; software, E.K. and M.M.; validation, J.J.v.T., M.P.A. and E.K.; formal analysis, M.M. and E.K.; investigation, M.M; resources, J.J.v.T. and E.K.; data curation, M.M.; writing—original draft preparation, M.M.; writing—review and editing, J.J.v.T. and M.P.A.; visualization, M.M.; supervision, J.J.v.T. and M.P.A.; project administration, J.J.v.T.; funding acquisition, J.J.v.T. All authors have read and agreed to the published version of the manuscript.

**Funding:** This research was funded by the National Research Fund (NRF) and South Africa—Mozambique—Zambia NRF Trilateral Joint research (ZAM180911357528-118479). This project was also partially funded by the Department of Environmental, Forestry and Fisheries (DEFF) under the National Resource Management (NRM).

**Acknowledgments:** The authors would like to acknowledge Chief LB Mopeli and the Monontsha Traditional Council for granting the permission for the water surveys; Tshiamo Setsipane for assistance in the collection and handling of the water samples.

**Conflicts of Interest:** The authors declare no conflict of interest.

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
