# Peer review of "Sensitivity and Calibration of the FT-IR Spectroscopy on Concentration of Heavy Metal Ions in River and Borehole Water Sources"

_applsci, doi:10.3390/app10217785_

Round 1
Reviewer 1 Report
Fourier transform infrared spectroscopy was applied to qualitatively and quanitively determination of heavy metal ions river and borehole water sources. Huge number of experimental data supports that FT-IR application is very suitable for determination the heavy metal ions. The manuscript contains very useful results and it worth for publication in Appl. Science. The paper is acceptable for publishing after minor modification.
Suggestions and remarks:
- The infrared spectroscopy serve very useful data in many field of science. I suggest expanding the presented field a little bit. Please mention in the Introduction that the applied method may use for characterization of heavy metal in ionic or sub-nanosized cluster form in colloidal dispersed system or metals dispersed on supports. The infrared position can give information for oxidation state and for morphology of dispersed clusters of heavy. Some suggested literatures: J. Phys. Chem. C 114, (2010) 169-178,; Surf. Sci. Rep. 71, (2016) 473-546, ; Chalcogenid Lett. 12, (2015) 333-338.
- Air was used as a background. Was it purified to eliminate the contaminations?
- Was the sample compartment purged?
Reviewer 2 Report
The paper reports the calibration of attenuated total reflectance Fourier transform infrared spectroscopy (ATR-FTIR) through partial least squares (PLS) and principal component analysis (PCA) for the detection of heavy metals (Ag, Cd, Cu, Pb and Zn) in water samples. The approach is then applied for the quantification of heavy metals in water samples from three rivers and four water sources. Results are compared with those from atomic absorption spectroscopy (AAS).
The work is not clearly written and conclusions are not sufficiently supported by data. The concentrations measured by means of the two chemometric procedures (PLS and PCA) are often different and different from the outcomes of AAS, thus questioning the reliability of the proposed FTIR methods. Data treatment and the adopted procedure are not adequately described, the discrepancies between the different concentrations measured by different methods are not convincingly explained and the English needs to be revised.
Overall, while I’m seriously convinced that FTIR spectroscopy can (and must) compete with analytical methods commonly employed for monitoring heavy metals (as well as other contaminants), the present paper does not provide any significant contribution to the field, rather it seems to discredit the technique in comparison to e.g. AAS. Hence, I regret that I cannot recommend the publication of this contribution.
Reviewer 3 Report
In this manuscript, the authors implemented Fourier transform infrared spectroscopy (FTIR) with partial least square (PLS) and principle component regression analysis (PCA) method to measure the concentration of heavy metal ions in river and water sources. The results are also compared with atomic absorption spectrometer (AAS) analysis. The authors concluded that FTIR has the potential for heavy metal analysis. However, there are several problems about the paper (see below). A major revision is needed.
1. In the abstract, line 19, PCA and PLS should not be re-defined. Line 18, AAS should be defined.
2. Paragraph starting from line 55 to 61 carries the same meaning as paragraph from line 42. These two paragraphs should be rewrite and combined.
3. Line 45, the authors say that “IR radiation excites the vibration of covalent bonds between atoms”, which is not entirely correct. The coordination bond in metal complexes and ionic bond in crystals are also IR active.
4. The major problem of the paper is that the authors used FTIR method but do not show any IR spectra. The IR spectra of all the heavy metals should be shown and key IR bands need to be assigned. See figure 3 and 4 and peak assignment in ref. 6. The authors said they collected the spectra in mid-infrared range (400-4000 cm-1).
5. What IR band are the authors attributed to the heavy metals? At which wavenumber? If it is just the heavy metal ion, then there is no IR band because it is just a single atom/ion. The heavy metals have to form some sort of complexes with either water or organic solvents in order to have a vibrational frequency.
6. The authors should explain figure 1 and 2 in more details. What are the number of principal component used and number of factors? Why sometimes the numbers are different? Where are the correction, cross-correction and ignore shown on the figures?
7. Line 93 to 98 are exactly the same as ref. 6. Although the authors used the same methods as ref.6, but they must not directly copy the equations and parameters. This must be avoided.
Reviewer 4 Report
The authors presented a paper on the application of the FTIR spectroscopy for the quantitative determination of heavy metal ions in water; the sensitivity and calibration of the method.
- The authors should present the FTIR spectra and mark the absorption bands that were taken into account for the evaluation of individual ions in the tested water samples
- FTIR spectra of the quantitative analysis of all important peaks in the tested water samples should be shown
- Full method validation should be presented including assessment of linearity, detection and quantification limits, precision, etc.
- Authors must further highlight the novelty of this work and why it is such important (in Introduction)
Reviewer 5 Report
- Page 2 lines 42-43: The authors refer to works in which IR spectroscopy was used in biological objects and oils. There are no references to published works in which this method was applied to aqueous samples.
- What bands in the ATR/FTIR spectra related to the metal ions in the liquid samples? Give an example of the spectrum of one of the samples.
- Section Discussion.
- Hard-to-read the text of the manuscript. Authors do not provide explanations for the models used. If we turn to the cited literature, then as far as I can understand, the authors offer two of the four used statistical models of data processing. To increase the readability of the article, it is necessary to present in more detail the initial IR spectra of the samples obtained and to justify the choice of models.
Round 2
Reviewer 2 Report
I truly appreciate the efforts made by the authors to improve their manuscript which is actually improved I several aspects.
Unfortunately, the main fundamental concern, i.e. the considerable number of significant differences between the concentrations measured through IR spectroscopy and AAS, still remains, questioning the reliability of the proposed approach. The motivations given for the reported discrepancies are not (fully) consistent and, in any case, different analytical methods cannot provide concentrations showing those differences.
Given this scientific issue, I really regret that I cannot recommend the paper for publication.
Reviewer 3 Report
The authors have addressed some of my concerns. There are a few more questions.
1. Line 120, the y(A) in equation should be y(C) based on line 121.
2. In section 3.1, the authors did not describe the IR spectra correctly. First of all, almost all the bands the authors observed are from water, such as OH stretching. They are not from the metals, adding the metal ions in will affect the water bands. The section title should be something like FTIR spectra of water containing metal ions. Second, the four peaks in table 1 need to be assigned, what are the corresponding vibrational motions?
3, In table 1, how are the peak positions with heavy metals determined? Based on figure 1, I cannot see any difference between these positions from different metals.
4. Can the authors determine the concentration of each heavy metals if they are mixed in water solution? What are the limitations?
5. Section 4, Discussion. Line 35-36, the authors discussed the IR band assignment in another paper, however, giving that the calibration solutions the authors prepared only contains water and metal ions, they should not see CH3/Ch2 symmetric stretching or C=O stretching. Again, the IR band in figure 1 need to be properly assigned.
6. Section 5, conclusion. Line 113 to 114, 3357 to 1635 are wavenumbers, not wavelengthes. Also, as I mentioned before, the authors are seeing the how the water bands are affected by adding these metals, not direct seeing these metals.
Reviewer 4 Report
Thank you for referring to my recommendations.
Author Response
Thank you so much for all the valuable and important contributions towards the improvement of this manuscript.
Round 3
Reviewer 2 Report
I don't believe pretending that the AAS data doesn't exist, then removing it, represents a good scientific way to tackle the problem..Author Response
Please see the attachment

Reviewer 3 Report
The authors have addressed my concerns and the manuscript should be published in its present form.
Author Response
Thank you for all the important and valuable contributions towards this manuscript.